# How Scarification, GA_3_ and Graphene Oxide Influence the In Vitro Establishment and Development of Strelitzia

**DOI:** 10.3390/plants12112142

**Published:** 2023-05-29

**Authors:** Patrícia Duarte de Oliveira Paiva, Diogo Pedrosa Correa da Silva, Bruna Raphaella da Silva, Israela Pimenta de Sousa, Renato Paiva, Michele Valquíria dos Reis

**Affiliations:** 1Departamento de Agricultura, Escola de Ciências Agrárias, Universidade Federal de Lavras, Lavras 37200-000, MG, Brazil; pedrosacorrea@yahoo.com.br (D.P.C.d.S.); michele.reis@ufla.br (M.V.d.R.); 2Departamento de Biologia, Instituto de Ciências Naturais, Universidade Federal de Lavras, Lavras 37200-000, MG, Brazil; bruna_rs@outlook.com.br (B.R.d.S.); israelapimenta@hotmail.com (I.P.d.S.); renpaiva@ufla.br (R.P.)

**Keywords:** *Strelitzia reginae*, dormancy break, nanomaterials, antioxidant system, in vitro propagation, bird-of-paradise

## Abstract

The propagation of strelitzia plants can be carried out in vitro as an alternative to combine the aseptic conditions of the culture medium with the use of strategies to promote germination and controlled abiotic conditions. However, this technique is still limited by the prolonged time and low percentage of seed germination, which is the most viable explant source, due to dormancy. Thus, the objective of this study was to evaluate the influence of chemical and physical scarification processes of seeds combined with gibberellic acid (GA_3_), as well as the effect of graphene oxide in the in vitro cultivation of strelitzia plants. Seeds were subjected to chemical scarification with sulfuric acid for different periods (10 to 60 min) and physical scarification (sandpaper), in addition to a control treatment without scarification. After disinfection, the seeds were inoculated in MS (Murashige and Skoog) medium with 30 g L^−1^ sucrose, 0.4 g L^−1^ PVPP (polyvinylpyrrolidone), 2.5 g L^−1^ Phytagel^®^, and GA_3_ at different concentrations. Growth data and antioxidant system responses were measured from the formed seedlings. In another experiment, the seeds were cultivated in vitro in the presence of graphene oxide at different concentrations. The results showed that the highest germination was observed in seeds scarified with sulfuric acid for 30 and 40 min, regardless of the addition of GA_3_. After 60 days of in vitro cultivation, physical scarification and scarification time with sulfuric acid promoted greater shoot and root length. The highest seedling survival was observed when the seeds were immersed for 30 min (86.66%) and 40 min (80%) in sulfuric acid without GA_3_. The concentration of 50 mg L^−1^ graphene oxide favored rhizome growth, while the concentration of 100 mg L^−1^ favored shoot growth. Regarding the biochemical data, the different concentrations did not influence MDA (Malondialdehyde) levels, but caused fluctuations in antioxidant enzyme activities.

## 1. Introduction

Strelitzia or bird-of-paradise (*Strelitzia reginae*) is a monocotyledonous plant endemic to South Africa with great horticultural potential due to its exotic-shaped flowers and the possibility of use in gardens and cut flower production [1].

Propagation of strelitzia is typically done through rhizome division and seeds. Rhizome utilization allows obtaining a reduced number of seedlings from a mother plant. Seed propagation is facilitated by the high number of dehiscent fruits produced by the inflorescence. However, there is a limitation due to the prolonged germination time caused by physical and chemical seed dormancy [1,2,3]. These propagation limitations highlight the need for studies to improve the commercial production process of seedlings, especially through seeds, by identifying efficient treatments to overcome dormancy and optimize germination [1,4].

An important alternative is the possibility of in vitro cultivation for the large-scale production of strelitzia seedlings, and some studies have already been conducted. However, research has shown the difficulty in germinating strelitzia seeds and embryos in vitro due to tegumentary and chemical dormancy [5], as well as oxidation occurrences [1,2].

Among the methods to overcome physical seed dormancy, scarification induces the breakage of dormancy, allowing for the exchange of water and gases between the seed components and the environment, facilitating embryo development. Chemical scarification is typically carried out with sulfuric acid, and physical scarification involves the use of sandpaper [4,6,7].

However, studies on dormancy break in strelitzia seeds are still scarce, and there is also an indication of the use of growth regulators [8,9,10], although most of them have not yet reached conclusive results.

To enhance the in vitro propagation process, studies have been conducted to evaluate the effect of temperature in the growth room on the formation of ROS (reactive oxygen species), and it was identified that increasing the temperature to 30 °C induces higher production of superoxide dismutase (SOD) and ascorbate peroxidase (APX) [11]. The cytogenetic stability during the in vitro cultivation process of strelitzia in media containing GA_3_ has also been analyzed, and stability was maintained [11]. In another study, the viability of cryopreservation of strelitzia zygotic embryos was investigated, achieving a survival rate of 66.13% for cryopreserved embryos after 30 min of dehydration [12]. Despite the studies conducted so far, there are still many gaps that need to be addressed to determine an efficient in vitro propagation process for strelitzia, such as determining an effective protocol for breaking seed dormancy and evaluating the effectiveness of adding GA_3_ to the culture medium to stimulate in vitro germination. Although the addition of GA_3_ has resulted in a high percentage of germination in strelitzia zygotic embryos [11], extracting the embryo is difficult and leads to a significant loss of seeds, highlighting the importance of determining an efficient protocol for germinating seeds.

The addition of GA_3_ to the culture medium aims to accelerate and uniformize seed germination [11,13]. Gibberellins act by weakening the endosperm layer that surrounds the embryo, facilitating the mobilization of endosperm reserves and activating vegetative growth [14].

In addition to the evaluation of conventional substances, the use of nanomaterials has emerged as an alternative in agricultural production, such as nanofertilizers, osmoprotectants, stress tolerance promoters, among others [15].

Nanomaterials (NMs) refer to all materials with dimensions ranging from 1 to 100 nm, possessing a large contact area due to their dimensions, which provides unique physicochemical properties such as high reactivity and absorption rates [16]. Each nanomaterial, with its own composition and molecular formula, can affect plants in different ways. Among the various materials, graphene oxide (GO), a carbonaceous nanomaterial consisting of a single layer of carbon derived from graphite sonication, stands out [17]. There are reports that this substance acts on plant tissues in a dose-dependent manner, with high concentrations causing toxicity and growth inhibition [18]. However, at ideal concentrations, GO has shown excellent results as a GO concentrations was observed to inhibit due to its hydrophilic character [19]. There are reports of its possible interaction with the auxin pathway, promoting synthesis and transport [20]. In addition, GO has been found to stimulate growth in Iris by enhancing photosynthetic performance [21].

Despite the growing advancements in research on GO and its effects, articles related to ornamental plants or in vitro cultivation are still scarce, with some studies focusing on major crops such as wheat and soybeans [20], fruit trees such as apple (*Malus domestica*) [22], leafy vegetables such as lettuce [23], and the model plant *Arabidopsis thaliana* [16].

In this context, the objective of this work was to evaluate how the dormancy breakage and in vitro germination of *Strelitzia reginae* seeds can be affected and improved through the use of chemical and physical scarification in combination with GA_3_, as well as the effect of graphene oxide.

## 2. Results and Discussion

### 2.1. Effect of Gibberellin Associated with Scarification on In Vitro Germination of Strelitzia Seeds

No interaction was observed between cultivation time, scarification, and the use of GA_3_ in seed germination. The highest germination rate, 53.57%, occurred after 30 days. After this period, a small but not significant increase was observed. Between 15 and 30 days, there was a difference of 8.49%, and after 45 and 60 days, there was an increase of 10.25% and 12.35%, respectively (Figure 1), demonstrating that germination up to 30 days should be considered, as there is no significant increase with time. Compared to the field germination time, 7 to 8 weeks [24], germination appears to be very favorable.

Chemical scarification was a crucial factor in increasing the germination percentage of the seeds, regardless of the use of GA_3_ (Figure 2).

Considering the immersion time on sulfuric acid, higher germination percentages were observed at 30 and 40 min. Seeds kept for longer periods, 50 or 60 min, did not show inhibition of the germination process, although the percentage was lower compared to the 30 and 40 min durations. However, it should be noted that the effectiveness of this scarification should be compared to the control without any treatment, or even a shorter duration of 10 min.

Seeds exposed to sulfuric acid for a short period of time may be inefficient in promoting germination, while prolonging the exposure period may affect seed viability [25], as observed in the case of strelitzia (Figure 2).

Studies on the ex vitro germination of strelitzia showed low germination percentages in seeds scarified with sulfuric acid. Approximately 38% germination was observed on paper towels when using 9 min of sulfuric acid treatment, and 47% germination was observed in a greenhouse with 7 min of immersion [4]. These ex vitro results highlight the need to use appropriate and longer treatment durations.

Based on the results, no effectiveness of GA_3_ in the germination process was observed. When comparing the best scarification durations, there was no difference in the different concentrations of gibberellic acid and the control. This indicates that scarification facilitates water absorption and embryo protrusion, and these processes occur independently of the presence of external concentrations of GA_3_.

Despite the results indicating that some interactions may have occurred regarding the use of GA_3_ and scarification periods, these are isolated and do not offer conclusive and assertive results proving their effectiveness.

The physical scarification of strelitzia seeds using sandpaper resulted in a germination percentage of 73.33% when supplemented with 27.71 mg L^−1^ of GA_3_ in the culture medium, which was higher compared to physical scarification used alone. However, seeds that were scarified with sandpaper and inoculated in the culture medium supplemented with lower concentrations of GA_3_ (6.93 and 13.85 mg L^−1^) or without the addition of GA_3_ showed a lower germination percentage (Figure 2). Therefore, to increase germination in strelitzia using physical scarification, the addition of GA_3_ at a concentration of 27.71 mg L^−1^ is essential.

Comparing chemical and physical scarification, a greater effectiveness of sulfuric acid can be observed compared to sandpaper scarification. The highest germination rate achieved with sandpaper scarification was 73.33% when 27.71 mg L^−1^ of GA_3_ was added. Immersion in sulfuric acid for 30 or 40 min resulted in average germination percentages of 81.66% and 93.33%, respectively, regardless of the addition of GA_3_. Thus, on average, there was an increase of 11.35% and 12.72% in seed germination, respectively. It is worth noting that sulfuric acid has limitations due to its hazardous manipulation and the production of toxic waste harmful to the environment. On the other hand, the use of sandpaper, although potentially more sustainable, is costlier, considering the small size of the seeds, the labor-intensive operation, and the need for the addition of GA_3_ to the medium, which also implies higher costs for the process.

Indeed, the use of GA_3_ was not necessary to break dormancy and increase germination, which contradicts previous research indicating its effectiveness in promoting strelitzia seed germination with the use of 750 mg L^−1^ of GA_3_ for 24 h, resulting in 78.10% germination on paper towels [9]. However, similar to what occurred in strelitzia, where the germination percentage was increased with scarification, the use of GA_3_ alone did not provide higher germination rates in *Tabernaemontana catharinensis*, as the dormancy of this species, according to the authors, may be due to the impermeable seed coat that was not altered by GA_3_ [26].

Each species may require a specific scarification period using sulfuric acid, which is the result of differences in seed coat thickness and chemical composition [27,28]. For example, in *Bowdichia virgilioides*, 5 min of immersion in sulfuric acid was sufficient to increase germination [29].

Analyzing the development of seedlings after 60 days of cultivation, the shoot length, root length, and number of leaves were also affected by scarification (Figure 3). Chemical scarification for 20, 30, and 40 min in sulfuric acid, as well as physical scarification, promoted the greatest shoot lengths. The shortest scarification times, 0 (control) and 10 min, as well as the longest scarification times, 50 and 60 min, resulted in lower values for shoot and root length. This can be explained by the lower germination of seeds observed in these treatments.

In the process of in vitro germination and development of strelitzia, the number of formed leaves was low, but they exhibited significant growth. It is observed that the number of leaves was less than 1 (Figure 4), and this value was independent of the interaction between scarification and the addition of GA_3_. The number of leaves, as well as the growth of the aerial part, was higher in seeds scarified for 20 to 40 min with sulfuric acid or using sandpaper, indicating the effectiveness of this process in accelerating germination and seedling development (Figure 4). The presence of GA_3_ resulted in a reduction in the number of leaves. GA_3_ influences plant growth by increasing both elongation and cell division, leading to increased internode distance, but it has no effect on the number of leaves [14].

Regarding the development of seedlings after 60 days, the interaction between scarification and GA_3_, as observed in germination, also played a role. The development of seedlings after 60 days of cultivation was greater in seeds that germinated with scarification for 30 and 40 min without GA_3_, with percentages of 86.66% and 80%, respectively, which did not differ significantly.

Another factor analyzed was survival, as in this species, germination can occur but the development of the seedling may not be completed. Thus, when scarification was performed for 30 min with the addition of GA_3_ to the medium, survival decreased by 40%, indicating that the effect of GA_3_ was not satisfactory. Similarly, for seeds scarified for 40 min with the addition of GA_3_ concentrations above 13.85 mg L^−1^, the percentage of development was also lower, reduced by 26.67% (Figure 5).

Sulfuric acid, when not used for the appropriate duration for each species, can have a detrimental effect. For strelitzia seeds, a shorter exposure time to sulfuric acid was not effective, while a longer exposure time was damaging.

Therefore, the most efficient method to increase the percentage of germination, seedling length, number of leaves, and survival is to perform scarification with sulfuric acid for 30 and 40 min (Figure 6B,C) compared to the control (Figure 6A). Scarification for 50 min (Figure 6D) can also be used to increase germination, but there is no need for longer exposure. With the use of chemical scarification, the addition of GA_3_ is unnecessary. The use of sandpaper for physical scarification promotes not only germination but also an increase in the number of leaves and seedling length. When combined with the addition of 27.71 mg L^−1^ GA_3_, the percentage of germination is increased (Figure 6E).

### 2.2. Analysis of Images of Strelitzia Seeds Subjected to Chemical and Physical Scarification

In the GroundEye^®^ analysis, the seeds showed variations in color, with black being predominant in all treatments (Figure 7a).

After acid scarification for 10 to 60 min, the percentage of black color decreased as the exposure time increased. As a result, chemically scarified seeds exhibited other colors, likely as a consequence of the acid’s effects or damage to the seed coat (Table 1).

The control treatment showed the highest percentage of black color (88.72%), with traces of sky blue (1.12%) and orange (0.74%). In seeds subjected to a 10 min scarification, which was insufficient to break dormancy, the seeds exhibited similar colorations compared to the control, but with a slight reduction in the percentages of sky blue and orange, and an increase in black color. Between 20 and 50 min, the period of greatest efficiency in chemical scarification, the black coloration showed higher percentages. Sky blue was not detected, only orange, compared to the control, but with the presence of red and traces of pink. With longer exposure to sulfuric acid, there was a reduction in black coloration and an increase in red and orange, along with the appearance of gray. This reduction in the percentage of black and the appearance of other colors confirm that seed dormancy in strelitzia is due to the presence of the seed coat. In seeds subjected to scarification by sanding, the color percentages were similar to the control, with the additional presence of light gray and cyan. The gray coloration appears to be the result of seed damage, considering that it occurred after physical scarification and also in seeds exposed for a long period in chemical scarification.

In a study on the characterization of strelitzia seed coloration, it was determined that black predominates, accounting for 80% of the seed, with lower values for blue (7%), sky blue (7%), and dark gray (6%) [3]. Since red coloration was not detected, it is evident that it is associated with the reaction resulting from the action of sulfuric acid, as well as the increase in orange percentages (Table 1).

In studies of exudates released in the culture medium during the in vitro cultivation of strelitzia, which caused oxidation, black, brown, lilac, and blue colors were observed [30]. By analyzing the infrared spectra of the compounds, the presence of a carbonyl group was identified in the lilac compound but not in the brown compound. The carbonyl group was present in secondary metabolites formed, probably due to environmental variations, plant age, and molecule degradation. Through the infrared spectra, it can be identified that the brown compound contains an OH group, characterizing it as a phenol or its derivative [30].

The spectra confirm that the brown and lilac samples correspond to different substances, and by overlaying some bands, it can be inferred that the brown and lilac colors represent dilutions or occur in lesser intensity of the black and blue colors, respectively [30].

In the radiographic analysis, it was observed that some seeds scarified for 60 min had cracks in the endosperm (Figure 7G). Scarification with sulfuric acid for 60 min damaged the seeds and impaired strelitzia germination, as the acid can penetrate the seed cells [25] and affect seed viability [31]. Some seeds subjected to sandpaper scarification showed possible damage to the embryo in the sanded area, which may have caused the unviability of some embryos, as severe physical scarification that causes injuries to the seed, especially the embryo, can lead to their unviability [6].

Based on the information provided, it can be inferred that the chemical dormancy exhibited in the seed coat is a result of phenolic compounds present, as indicated by the high percentage of black coloration in the seeds.

### 2.3. Effect of Graphene Oxide on the In Vitro Growth of Strelitzia Seedlings

The tested concentrations of graphene oxide (GO) influenced the aerial part, plant weight, rhizome length, and diameter. For the aerial part, it promoted greater growth compared to the other concentrations, which were similar to the control or lower, such as the use of 200 mg L^−1^, indicating an inhibitory effect (Figure 8A). Similarly, the plant weight was higher at a concentration of 100 mg L^−1^, as a result of the increased growth (Figure 8B).

Analyzing the rhizome length, the concentration of 100 mg L^−1^, contrary to its effect on the aerial part, resulted in lower rhizome growth (Figure 8C). However, it exhibited a larger diameter (Figure 8D). GO has been reported to act in a dose-dependent manner [18], and this behavior was observed in strelitzia. At lower concentrations (10 and 50 mg L^−1^), both aboveground and rhizome growth tended to be similar to the control, but the higher concentration, 200 mg L^−1^, appeared to inhibit the development of the seedlings. At low concentrations, graphene oxide can stimulate plant growth, increase the activity of antioxidant enzymes, and improve photosynthetic efficiency. This can be explained by the fact that graphene oxide can act as a stabilizer of cell membranes, enhancing plant resistance to oxidative stress [21,32].

At the higher concentration (200 mg L^−1^), there was a reduction in biomass accumulation and rhizome and aboveground length, and a decrease in rhizome diameter. In the absence and at a low concentration (10 mg L^−1^) of GO, the effects observed were similar (Figure 8). Studies show that different concentrations of GO have different biological effects on plants, ranging from growth stimulation to toxicity [32]. In different rice genotypes, the effect of increasing GO concentrations was observed to inhibit aboveground elongation [33]. On the other hand, in wheat seedlings, increasing concentrations favored both shoot and root elongation [34].

At high concentrations, graphene oxide can be toxic to plants, causing growth inhibition, decreased antioxidant activity, and alteration in the expression of genes involved in metabolism. This can occur due to excessive oxidative stress generation, leading to cellular damage and cell death [35]. Regarding the number of leaves and number of rhizomes, the obtained values were not statistically significant, as the seedlings on average had 0.87 leaves and 1.08 rhizomes. In this way, based on the presented results, the concentration of 100 mg of GO demonstrated to be more effective for strelitzia.

### 2.4. Biochemical Analysis

The quantification of hydrogen peroxide and lipid peroxidation, as well as the determination of POD, SOD, and CAT activity and protein quantification, showed results that indicated an interaction between the cultivation time and the concentrations of GO (Figure 9).

Analyzing the levels of H_2_O_2_, after 7 days, they were significantly reduced with increasing concentrations of GO compared to the control. It can be observed that as the seedlings grew, the levels of H_2_O_2_ decreased, according to the analyses conducted at 14 and 21 days. However, at the end of the period, when the seedlings were well developed, H_2_O_2_ tended to increase, with higher values observed in the control treatment without the addition of GO (Figure 9A). H_2_O_2_ production seems to be connected to the physiology of development, especially during root formation [36,37].

Regarding the levels of malondialdehyde (MDA), a product of lipid peroxidation, there was no significant interaction between the evaluation times and GO concentrations. MDA levels showed a reduction after 14 days, coinciding with the development of the seedlings (Figure 9B). The accumulation of MDA in plants can negatively affect root growth, seed germination, photosynthesis, nutrient uptake, and enzyme activity. Moreover, MDA can interfere with gene expression and hormonal regulation, affecting plant responses to abiotic and biotic stresses [38]. The highest initial value was observed at 7 days, corresponding to the initial phase of germination with radicle protrusion. During the germination process, there is a high reactivation of respiratory metabolism, with a high presence of reactive oxygen species and lipid peroxidation related to the energy demand for germination [39].

For the specific activity of the peroxidase enzyme (POD), the activity was higher at 7 days, the initial germination period, and decreased afterwards with seedling development. Additionally, the activity was higher in the control treatment compared to the seedlings from the treatments with GO. This indicates the influence of GO in controlling the specific activity of POD over time.

Analyzing the superoxide dismutase (SOD) activity, an increase in enzyme activity was observed over time (Figure 9D). A similar pattern was observed for CAT activity (Figure 9E). Higher CAT enzyme activity under in vitro conditions may be related to the generation of H_2_O_2_ as a signaling molecule in morphogenic responses [12,40].

Regarding the protein content in the rhizomes, the levels showed a reduction over time, both in the treatments with GO and in the control, indicating that the addition of GO to the medium does not alter development (Figure 9F).

A possible explanation for the increase in SOD activity at 21 days in the concentration of 100 mg L^−1^ compared to the others is that SOD acts in the first line of defense against superoxide anions, and the increase in its activity indicates an elevation of these anions, which, similar to hydrogen peroxide, can cause oxidative stress [37]. This peak coincided with an increase in CAT activity, working together with SOD. CAT acts in the second line of defense, catalyzing the breakdown of hydrogen peroxide. The same peak in activity is observed at 28 days in the absence of GO.

There are several enzymes that can degrade malondialdehyde (MDA), a product of lipid peroxidation, in plants. Antioxidant enzymes such as superoxide dismutase (SOD), catalase (CAT), and peroxidase (POD) play an important role in protecting plant cells against oxidative stress [12,37,41].

SOD is an enzyme that catalyzes the conversion of superoxide into hydrogen peroxide (H_2_O_2_), which can be decomposed by CAT into water and oxygen. POD is another antioxidant enzyme that utilizes H_2_O_2_ to oxidize a wide range of substrates, including MDA [42].

In general, the concentration of 100 mg L^−1^ OG resulted in a reduction in hydrogen peroxide at 7 and 21 days (Figure 9A). This reduction was attributed to the increased activity of the POD enzyme at 7 days (Figure 9C), and the SOD and CAT enzymes at 21 days (Figure 9D,E), anticipating the peaks of enzyme activation by 7 days. This favored the reduction in the production of ROS in in vitro-cultivated strelitzia seedlings.

## 3. Material and Methods

### 3.1. Effect of Gibberellin Associated with Scarification on In Vitro Germination of Strelitzia Seeds

Mature strelitzia (*Strelitzia reginae*) seeds were collected and had their arils manually removed before undergoing both physical and chemical scarification.

Chemical scarification: The seeds were immersed in concentrated sulfuric acid (P.A.) for 10, 20, 30, 40, 50, and 60 min, followed by triple rinsing with distilled water.

Physical scarification: The seeds were manually sanded on the opposite side of the radicle protrusion (micropyle). Afterward, they were soaked in distilled water for 120 min for pre-hydration.

Untreated seeds were used as control. For these analyses, 15 replications per treatment were used, with each replication representing one seed per test tube (plot).

#### 3.1.1. Effect of Gibberellin Associated with Seed Scarification on In Vitro Germination

After scarification (chemical and physical), the seeds were disinfested with 70% ethanol for 30 s and 2.5% sodium hypochlorite for 15 min. They were then rinsed three times with distilled and autoclaved water (modified from [5]). The seeds were inoculated on MS medium [24] supplemented with 30 g L^−1^ sucrose, 0.4 g L^−1^ PVP, 2.5 g L^−1^ Phytagel^®^ (Sigma, U.S.) [5], and the following treatments: GA_3_ at concentrations of 0 (control), 6.93, 13.85, and 27.71 mg L^−1^. The pH was adjusted to 5.8 before autoclaving at 121 °C for 20 min.

After inoculation, the seeds were kept in the dark for 7 days and then transferred to a growth room with a photoperiod of 16 h, a temperature of 25 ± 2 °C, and a photon irradiance of 36 μmol m^−2^ s^−1^.

The percentage of germination was evaluated at 15, 30, 45, and 60 days after inoculation. At 60 days, the percentage of survival, shoot length (cm), main root length (cm), and number of leaves were assessed.

#### 3.1.2. Characterization of Seed Integrity after the Process of Chemical and Physical Scarification

The scarified seeds (chemical and physical) were characterized using images obtained with the GroundEye^®^ (Tbit Tecnologia S.A., Lavras-MG, Brazil) system and radiographic analysis to assess the integrity of the seed coat at different times of chemical and physical scarification, as well as the control group. The seeds were placed in the tray of the GroundEye^®^ v. S120 image capture device. Analysis configuration was then performed to calibrate the background colors using the CIELab color model, with a luminosity index ranging from 0 to 100, a dimension a ranging from −15.6 to 44.4, and a dimension b ranging from −58.7 to −18.7. After calibrating the background color, the images were analyzed, and color dominance characteristics were extracted.

For radiographic analysis, the Faxitron HP^®^ equipment, Model 43855AX (USA), with Kodak Min-R 2000 radiographic film measuring 18 × 24 cm, was used. X-ray processing was performed using a Kodak brand X-ray processing processor (USA), Model M35X OMAT. To assess seed viability, the seeds were fixed on transparent acetate sheets using double-sided tape. A radiation intensity of 35 kV and an exposure time of 19 s were used for equipment calibration.

### 3.2. Effect of Graphene Oxide on In Vitro Culture

After the collection of mature seeds, the aril was removed to avoid the contamination of the material cultivated in vitro, and then chemical scarification was performed using sulfuric acid (H_2_SO_4_) for 30 min. For asepsis, the seeds were taken to the laminar flow, immersed in 70% alcohol for 30 s, and then rinsed three times in autoclaved distilled water (121 °C for 20 min). After that, the seeds were immersed in 2.5% (*w*/*v*) sodium hypochlorite (NaCl) for 15 min and rinsed again in autoclaved distilled water [5].

The seeds were inoculated on MS culture medium [43], supplemented with 30 g L^−1^ sucrose, and solidified with 2.5 g L^−1^ Phytagel^®^. The treatments tested consisted of different concentrations of graphene oxide (GO): 0 (control), 10, 50, 100, and 200 mg L^−1^ added to the culture medium. The pH of the culture medium was adjusted to 5.8 [19,44]. Sterilization of the culture medium was performed by adding 1.0 mL L^−1^ of the sanitizing agent DIOXIPLUS^®^ (Dioxide, Indaiatuba-SP, Brazil). Twenty strelitzia seeds were inoculated per treatment, totaling the replicates, with one seed per test tube (plot). After 30 days, the length of the shoot (cm), length of the root system (cm), number of leaves, number of roots, fresh weight (g), root diameter, and leaf area (cm) were evaluated, in addition to performing biochemical analyses of the seedlings.

#### 3.2.1. Biochemical Analysis

For the biochemical analyses, strelitzia rhizomes obtained from in vitro cultivated seedlings were used at concentrations of 0 (control), 50, and 100 mg L^−1^ of graphene oxide. Sampling for each concentration was conducted on days 7, 14, 21, and 28 of cultivation.

#### 3.2.2. Quantification of Hydrogen Peroxide (H_2_O_2_) and Lipid Peroxidation

The rhizomes were macerated using liquid nitrogen (LN) with 20% polyvinylpolypyrrolidone (PVPP) (*w*/*v*) and subsequently homogenized in 0.1% (*w*/*v*) trichloroacetic acid (TCA). The macerate was then centrifuged at 12,000× *g* for 15 min at 4 °C, and the supernatant was collected and stored in an ultrafreezer at −80 °C [45]. Hydrogen peroxide analysis was conducted following the method described by [45]. A 45 µL aliquot of the supernatant was combined with 180 µL of the stock solution containing 10 nM potassium phosphate buffer at pH 7 and 1 M potassium iodide. The plates were read at 390 nm using a spectrophotometer. Lipid peroxidation quantification was performed using the TBARS method proposed by [46]. A 125 µL aliquot of the supernatant was added to a reaction mixture containing 250 µL of 0.5% thiobarbituric acid (TBA) and 10% trichloroacetic acid (TCA), followed by incubation at 95 °C for 30 min. The samples were rapidly cooled using ice, and the plates were read at 535–600 nm using an Elisa (LMR-96, Loccus, São Paulo-SP, Brazil). Malondialdehyde (MDA) is a low-molecular-weight end product formed by the decomposition of lipid peroxidation product.

#### 3.2.3. Antioxidant Enzyme Activity: Catalase (CAT), Peroxidase (POD), and Superoxide Dismutase (SOD)

Extraction of antioxidant enzymes was obtained by macerating 100 mg of rhizomes in liquid nitrogen and PVPP, followed by addition to 1.5 mL of extraction buffer containing: potassium phosphate buffer 400 mM, pH 7.8; EDTA 10 mM; ascorbic acid 200 mM; and water. The extract was then centrifuged at 13,000× *g* for 10 min at 4 °C, and the supernatant was collected and stored at −20 °C until analysis. The supernatants were used for the quantification of catalase (CAT), superoxide dismutase (SOD), and peroxidase (POD) activity [47].

CAT activity was determined using the method of [48]. An aliquot of 9 µL was added to 163 µL of incubation buffer containing: potassium phosphate buffer 200 mM, pH 7.0; water; and hydrogen peroxide, 250 mM, at 30 °C. Plate readings were taken at 240 nm every 15 s for 3 min. The molar extinction coefficient of 36 nM^−1^ cm^−1^ was used.

POD activity was evaluated by pipetting the supernatant obtained from centrifugation (4 µL) and adding 163 µL of incubation buffer [100 µL sodium phosphate buffer 100 mM (pH 6.0), 33 µL guaiacol 0.8%, and 30 µL water] at 25 °C. Then, 33 µL of H_2_O_2_ (0.9%) was added. Absorbance measurement was performed at 470 nm using a spectrophotometer for 3 min [49].

SOD activity was evaluated based on the enzyme’s ability to inhibit the photoreduction of nitrotetrazolium blue (NBT). The enzyme extract (10 µL) was added to 190 µL of incubation buffer containing: phosphate buffer 100 mM, pH 7.8; methionine 70 nM; EDTA 10 µM; water; NBT 1 mM; and riboflavin 0.2 mM. The plates with the samples and incubation buffer were illuminated with a 20 W fluorescent lamp for 7 min, and readings were taken at 560 nm using a spectrophotometer [50].

### 3.3. Statistical Analysis

All experiments were set up according to a completely randomized design (CRD). The data obtained from the experimentation were subjected to analysis of variance using the statistical software SISVAR^®^ (UFLA, Version 5.6), and the means were compared using the Tukey test at a 5% probability level [51].

## 4. Conclusions

The use of chemical scarification with sulfuric acid for a period of 30–40 min promotes higher germination of in vitro strelitzia seeds, greater seedling growth, and increased survival, making it an important alternative for propagating this species.

In strelitzia, the presence of GO in the culture medium acts in a dose-dependent manner. At low concentrations, it is not effective, and at high concentrations, it affects development. The concentration of 100 mg L^−1^ GO favored the in vitro growth and development of strelitzia seedlings, and the responses related to the plant’s antioxidant system were evident.

The use of 100 mg L^−1^ GO in the culture medium of in vitro-cultivated strelitzia seedlings resulted in a reduction in the amount of hydrogen peroxide through the activation of the enzymes POD, CAT, and SOD.

## Figures and Tables

**Figure 1 plants-12-02142-f001:**
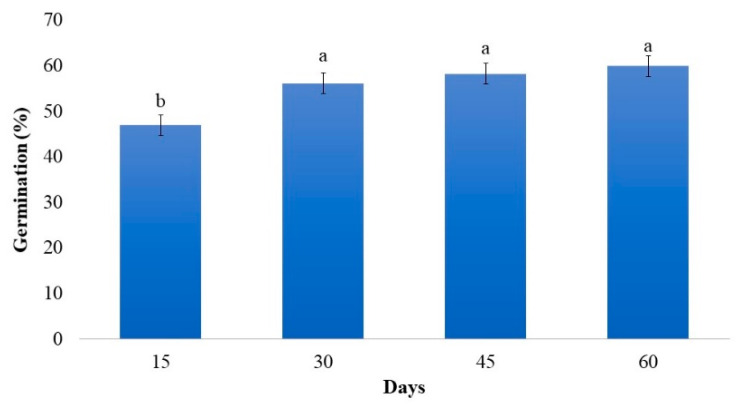
Germination percentage in vitro of strelitzia seeds as a function of cultivation time. Data are expressed as mean ± standard error (bars). Means followed by the same letter do not differ significantly from each other according to the Scott–Knott test (*p* < 0.05).

**Figure 2 plants-12-02142-f002:**
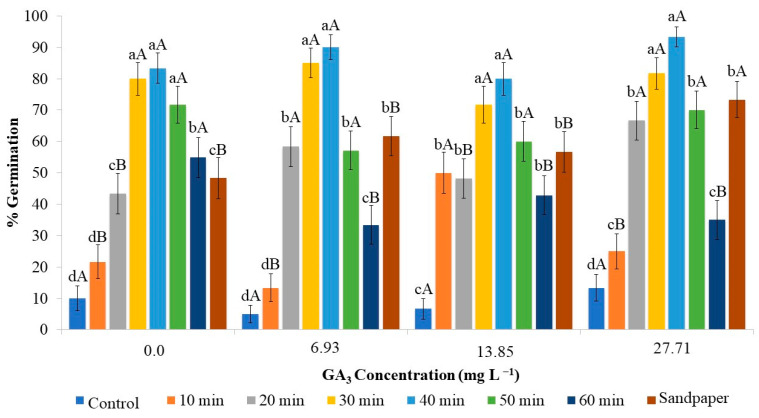
In vitro germination percentage of strelitzia seed in 30 days as a function of chemical and physical scarification of seeds and different concentrations of GA_3_. The data are expressed as mean ± standard error (bars). Means followed by the same lowercase letter do not differ from each other for seed scarification time, and means followed by the same uppercase letter do not differ from each other for GA_3_ concentration according to Scott–Knott test (*p* < 0.05).

**Figure 3 plants-12-02142-f003:**
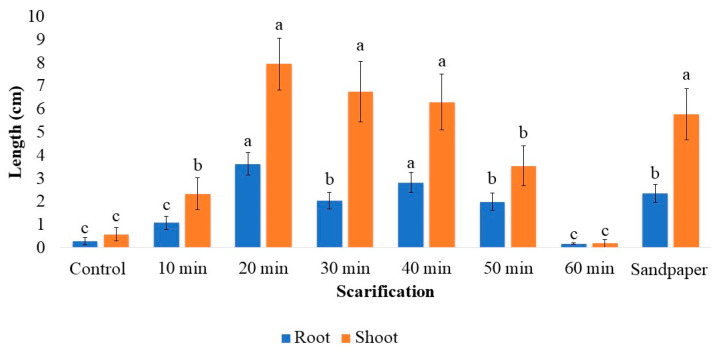
Average shoot length (orange) and length of the longest root (blue) at 60 days of in vitro cultivation as a function of chemical and physical scarification of strelitzia seeds. Data are expressed as mean ± standard error (bars). Means followed by the same letter do not differ significantly according to the Scott–Knott test (*p* < 0.05).

**Figure 4 plants-12-02142-f004:**
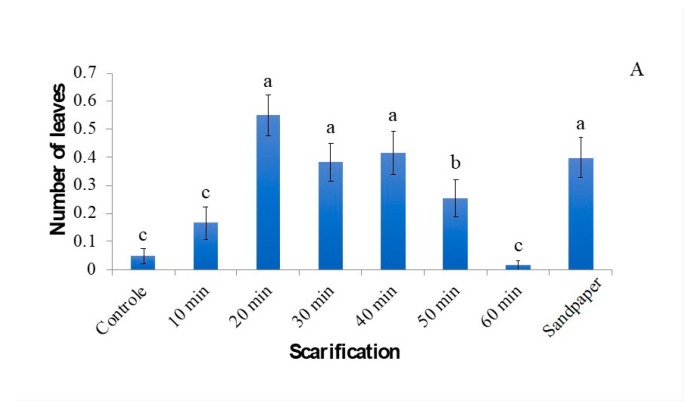
Number of leaves (average) of strelitzia in different concentrations of GA_3_ (**A**) and chemical and physical scarification of the seeds (**B**) after 60 days of in vitro cultivation. The data are expressed as mean ± standard error (bars). Means followed by the same letter do not differ significantly according to the Scott–Knott test (*p* < 0.05).

**Figure 5 plants-12-02142-f005:**
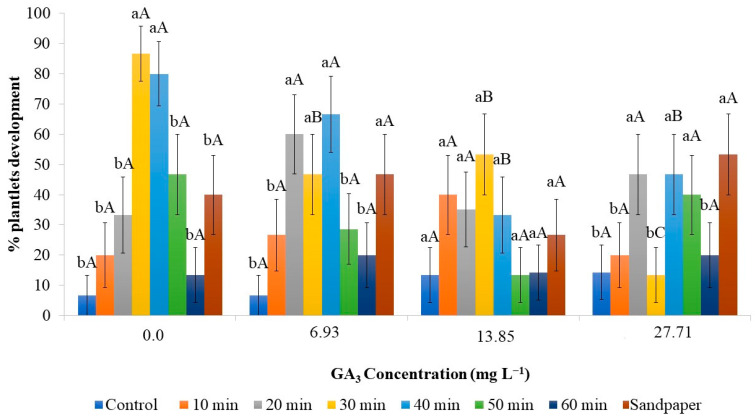
Percentage of seedling development of strelitzia as a function of chemical and physical scarification and different concentrations of GA_3_ after 60 days of in vitro cultivation. The data are expressed as mean ± standard error (bars). Means followed by the same lowercase letter do not differ significantly for seed scarification, and means followed by the same uppercase letter do not differ significantly for GA_3_ concentration according to the Scott–Knott test (*p* < 0.05).

**Figure 6 plants-12-02142-f006:**
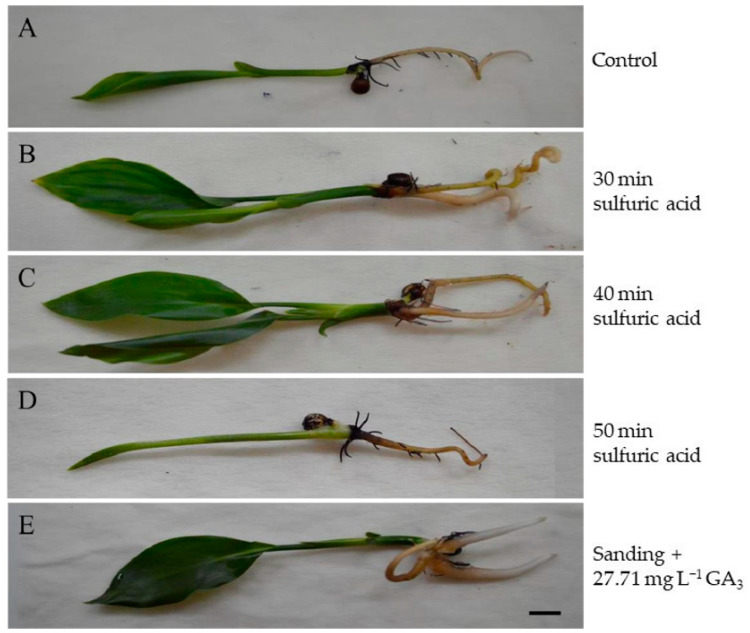
Strelitzia seedlings at 60 days of in vitro cultivation. Scale bar: 1 cm.

**Figure 7 plants-12-02142-f007:**
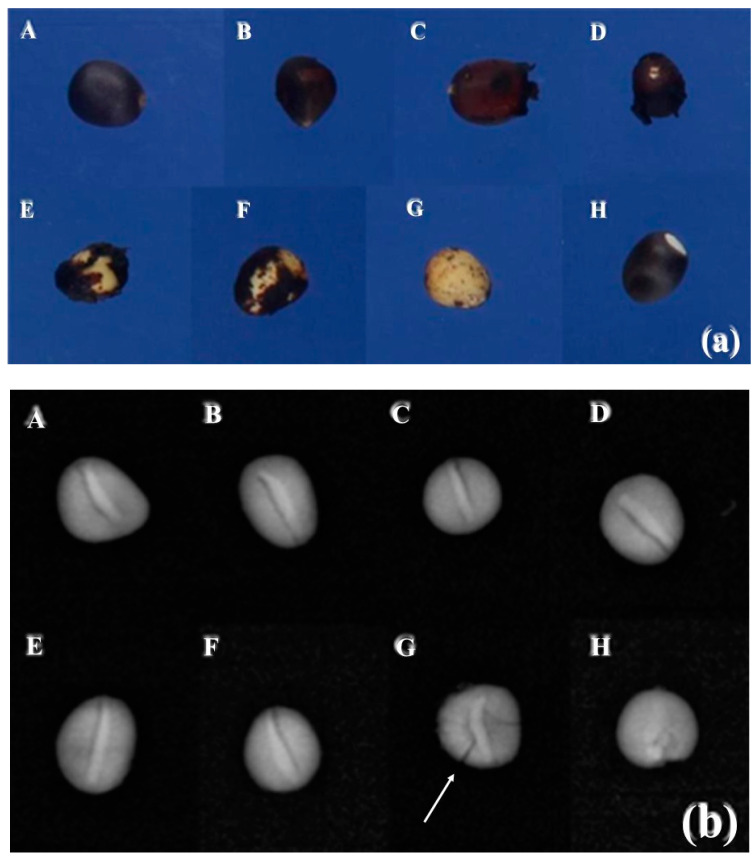
Images of strelitzia seeds obtained by (**a**) GroundEye^®^ and (**b**) X-ray device. (**A**) Untreated seeds. (**B**) Seeds scarified with sulfuric acid for 10 min. (**C**) Seeds scarified with sulfuric acid for 20 min. (**D**) Seeds scarified with sulfuric acid for 30 min. (**E**) Seeds scarified with sulfuric acid for 40 min. (**F**) Seeds scarified with sulfuric acid for 50 min. (**G**) Seeds scarified with sulfuric acid for 60 min. (**H**) Seeds scarified with sandpaper. Arrow points to the crack in the seed endosperm.

**Figure 8 plants-12-02142-f008:**
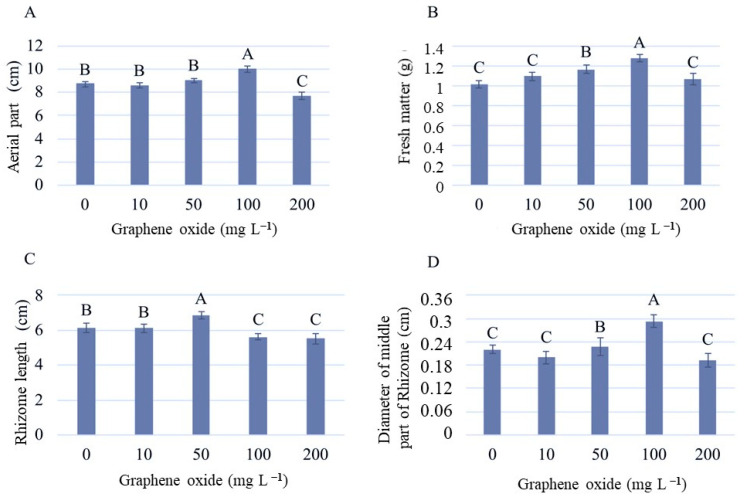
Aerial part (**A**), fresh matter (**B**), rhizome length (**C**), diameter of the middle portion of the rhizome (**D**) from different concentrations of graphene oxide (GO) in vitro. Uppercase letters that are the same do not differ from each other regarding the concentration of GO.

**Figure 9 plants-12-02142-f009:**
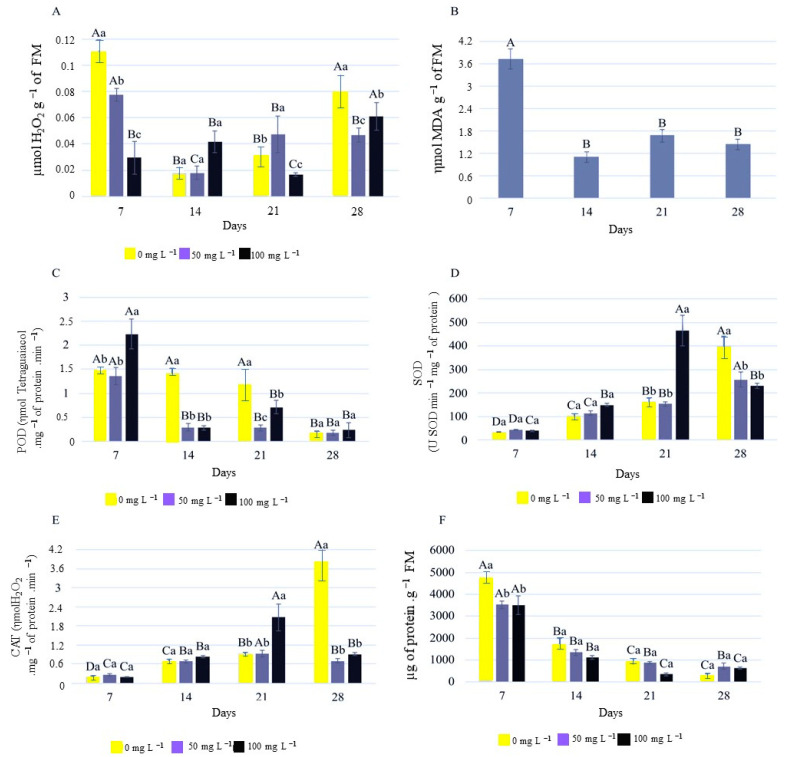
Hydrogen peroxide (**A**), lipid peroxidation (**B**), POD activity (**C**), SOD activity (**D**), CAT activity (**E**), and protein quantification (**F**) in in vitro cultivation of strelitzia in the presence of GO. Capital letters that are the same do not differ statistically in relation to the days. Lowercase letters that are the same do not differ in relation to the concentrations of GO.

**Table 1 plants-12-02142-t001:** Percentage of color variation in strelitzia seeds obtained from GroundEye^®^.

Color	Scarification							
	Control	10 min	20 min	30 min	40 min	50 min	60 min	Sand
Black	88.72	98.89	98.88	98.41	95.24	76.51	59.59	83.42
Sky blue	1.12	0.07	0	0	0	0	0	1.32
Gray (light)	0	0	0	0	0	0.05	0.13	2.56
Red	0	0	0.48	0.47	0.58	0.96	1.86	0
Orange	0.74	0.26	0.15	0.78	2.23	17.32	30.28	1.38
Pink	0	0	0.16	0.05	0	0	0	0
Cyan	0	0	0	0	0	0	0	0.04

## Data Availability

Data is contained within the article.

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
