# Peer review of "How Scarification, GA3 and Graphene Oxide Influence the In Vitro Establishment and Development of Strelitzia"

_plants, 2023, doi:10.3390/plants12112142_

Round 1

Reviewer 1 Report

In the “Abstract” section, please explain the abbreviations when they are used first time.

  In the penultimate paragraph, graphene oxide is incorrectly abbreviated.

For all the instruments used, please add city and country of origin.

In section 2.2 it is not very clearly how treatment with GO (Once again abbreviated incorrectly) was performed. GO was added to culture medium? Please explain.

There is many data in figure 2 and maybe it would be easier to read if the authors used colors instead of gray tones.

What represent Figure 3? Is it the sum of shoot and root plant length for all germinated plant?

It is normally than when are more germinated plant the sum to be higher. Maybe the authors should present the shoot and root plant length for the same number of plants not for every germinated plants.

The same for figure 4.

In Subsection 3.3 “Effect of graphene oxide on the in vitro growth of bird of paradise seedlings” It is not clear which concentration of GO had a good influence on growth of aerial part.

Please rephrase, some phrases do not have the verb, or are unclear to the readers. 

Author Response

In the “Abstract” section, please explain the abbreviations when they are used first time.

Okay

For all the instruments used, please add city and country of origin City?

Okay, for the ones possible.

In section 2.2 it is not very clearly how treatment with GO (Once again abbreviated incorrectly) was performed. GO was added to culture medium? Please explain.

Okay

There are many data in figure 2 and maybe it would be easier to read if the authors used colors instead of gray tones.

Okay, we have changed the figures

What represent Figure 3? Is it the sum of shoot and root plant length for all germinated plant?

Orange bars = shoot; Blue bars = root.

We have changed the caption text.

It is normally than when are more germinated plant the sum to be higher. Maybe the authors should present the shoot and root plant length for the same number of plants not for every germinated plant.

The values correspond to the average, not to a sum. Strelitzia plants grow fast and are higher.

In Subsection 3.3 “Effect of graphene oxide on the in vitro growth of bird of paradise seedlings” It is not clear which concentration of GO had a good influence on the growth of the aerial part.

We have inserted a new paragraph.

Author Response

Seeds were subjected to chemical scarification with sulfuric acid for different- inserttime periods and physical scarification (sandpaper), in addition to a control treatment without scarification.

Okay

with 30 g L-1 sucrose, 0.4 g L-1 PVP, 2.5 g L-1 Phytagel®, and gibberellic acid (GA3) at different concentrations. – consider changing -1 for gL as a superscript for the entire writeup

Okay

  1. INTRODUCTION ………………. room and the addition of GA3 ……. consider providing the full definition of acronym the very first time it is introduced. For example, what does GA3 stand for in full? We changed the text

Despite the growing advancements in research on OG….? What is OG?

GO – we have modified.

Figure2, Figure5: Please consider using different colors for the different treatments instead of the different shades of black/gray. As they are, it is not so easy to distinguish them.

Okay, we have modified.

However, seeds that were scarified with sandpaper and inoculated in the culture medium supplemented with lower concentrations of GA3……………………………is it GA3 or GA3? Check out for this for the rest of the article.

Okay, we have modified the text.

……. bird-of-paradise…… or bird of paradise……...please consider consistency in use of hyphenated words.

We have modified in the entire text to “strelitzia”